# Spatial-temporal assessment of future population exposure to compound extreme precipitation-high temperature events across China

Ke Jin[1,2,3], Yanjuan Wu[1,2,3], Xiaolin Sun[1,2,3], Yanwei Sun[1,2,3], Chao Gao[1,2,3]*

1 Department of Geography and Spatial Information Techniques, Ningbo University, Ningbo, China,
2 Zhejiang Collaborative Innovation Center & Ningbo Universities Collaborative Innovation Center for Land and Marine Spatial Utilization and Governance Research, Ningbo, China, 3 Donghai Academy, Ningbo University, Ningbo, China

* gaoqinchao1@163.com

**Data Availability Statement:** The original data that support the findings of this study are openly available. The population data provided by

## Abstract

Global warming has increased the probability of extreme climate events, with compound extreme events having more severe impacts on socioeconomics and the environment than individual extremes. Utilizing the Coupled Model Intercomparison Project Phase 6 (CMIP6), we predicted the spatiotemporal variations of compound extreme precipitation-high temperature events in China under three Shared Socioeconomic Pathways (SSPs) across two future periods, and analyzed the changes in exposed populations and identified influencing factors. From the result, we can see that, the CMIP6 effectively reproduces precipitation patterns but exhibits biases. The frequency of compound event rises across SSPs, especially under high radiative forcing, with a stronger long-term upward trend. Furthermore, the economically developed areas, notably China's southeastern coast and North China Plain, will be hotspots for frequent and intense compound extreme events, while other regions will see reduced exposure. Finally, in the long-term future (2070–2100), there is a noteworthy shift in population exposure to compound events, emphasizing the increasing influence of population factors over climate factors. This highlights the growing importance of interactions between population and climate in shaping exposure patterns.

## Introduction

In accordance with the Sixth Assessment Report (AR6) from the Intergovernmental Panel on Climate Change (IPCC), the average global surface temperature increased by 0.99°C during the initial two decades of the 21st century (2001–2020), as measured against the pre-industrial period [1]. Simultaneously, the World Meteorological Organization (WMO) notes that the past seven years (2015–2021) stand out as the warmest in recorded history [2]. This period is marked by a global five-year average temperature rise of 1.17 ± 0.13°C from 2018 to 2022 relative to the pre-industrial period [3]. Clearly, global warming has become an undeniable fact.

Professor Jiang Tong's team from the NUIST, which could also be obtained from the Science Data Bank (DOI: https://doi.org/10.57760/sciencedb.01683). The CMIP6 datasets can be found at https://esgf-node.llnl.gov/projects/cmip6. All processed and calculated data are within the article and its Supporting Information files.

**Funding:** The authors received no specific funding for this work.

**Competing interests:** The authors have declared that no competing interests exist.

This climate change, distinguished by the signature of global warming, has intensified the occurrence, frequency, duration, and spatial extent of extreme climate events, including heatwaves, extreme precipitation, droughts, floods, and typhoons [4]. Among these extreme events, extreme precipitation, high-temperature events, and their compound events emerge as globally perilous climatic disasters. Their hazardous nature is anticipated to intensify further in the future [5, 6]. This study delves into the increased intensity, frequency, duration, and spatial impact of these events. Among these extremes, compound events stand out as one of the most hazardous climatic disasters globally, and projections suggest a continued escalation in the future [7, 8].

As reported by the World Meteorological Organization [9] from 1970 to 2019, over 11,000 disasters related to weather, climate, and precipitation occurred globally. These events resulted in a cumulative death toll exceeding 2 million and economic losses amounting to $36.4 trillion. Global warming has altered the interplay of driving factors behind extreme weather, increasing the likelihood of concurrent or sequential extreme events [10]. Events occurring simultaneously or sequentially, often termed compound extremes, generally exhibit greater destructiveness than individual extreme events. They pose unprecedented threats and exert larger negative impacts on various systems or sectors, including water resource supply, agricultural production, public health, and infrastructure [11, 12]. Notable instances include the flood-heatwave compound event that struck western Japan in the summer of 2018, causing thousands of fatalities within a week [13, 14]. Another example is the compound event in February 2019 in Queensland, Australia, involving low temperatures, strong winds, and extreme precipitation, leading to the death of 500,000 cattle [15]. In the U.S. Midwest, extreme precipitation following summer heatwaves frequently triggers floods, posing significant challenges to food security and ultimately threatening human livelihoods [16]. Research has explored the probability and evolving characteristics of compound heatwaves and floods, revealing that after shorter and hotter heatwaves, heavy rainfall is more likely.

China emerges as a region highly responsive to and markedly influenced by global climate shifts. Over the period from 1951 to 2020, there was an increase of 0.26˚C in the annual average temperature per decade, surpassing the global average of 0.15˚C per decade [17]. Regarding precipitation, from 1961 to 2020, there is a subtle increasing trend in China's average annual precipitation. However, this increase is more pronounced in the context of extreme events. Since the mid-1990s, extreme high-temperature and intense precipitation events in China have markedly escalated, featuring greater intensity, broader impact, and distinct regional variations [18]. The economic ramifications of these extreme events in China, equivalent to 1.07% of the GDP annually, far exceed the global loss level of 0.14% [19]. Moreover, China stands out as one of the countries with the highest frequency of compound extreme events. The nation has undergone a swift process of urbanization since the 1970s [20]. This process has further intensified the characteristics of extreme weather and climate events across many regions, including changes in frequency, duration, and intensity [21, 22]. An imperative is placed on the investigation and evaluation of compound extreme events. This is essential to better assess and mitigate the potential impacts and risks arising from the expanding societal footprint on the environment.

Attention have been paid to this issue, some studies found that future drought-flood, water-flood, and drought-flood alternation events in China will increase significantly [23]. It was pointed out that the frequency of compound extreme drought and heatwave (CDHEs) in China in the future will be much higher than that of other compound events [24]. In addition, not only extreme compound events but also mild weather events have received attention. In a warmer future, the occurrence of mild weather in southeastern China will decrease

significantly compared to current levels. At the seasonal scale, the decrease in mild weather in summer outweighs the increase in spring and autumn [25].

Human-induced climate change has substantially impacted regions worldwide. Extensive research has been conducted on the evolving trends of individual extreme events globally [26–28]. However, the understanding of how compound extreme events, particularly extreme precipitation-heat combinations, vary over time and space in China remains limited. This study employs data from the CMIP6 and population grids based on the SSPs. By systematically comparing multiple model outputs with historical data, the optimal CMIP6 model is identified. Subsequently, this model is employed to analyze the temporal and spatial changes in compound extreme events, focusing on extreme precipitation and high temperatures, in China and its climatic sub-regions for the near-term future (2020–2050) and long-term future (2070–2100). The study also delves into the characteristics of population exposure to compound extreme events and associated influencing factors.

The focus of this investigation is on China and its sub-regions, where we scrutinize the spatiotemporal variances in compound extreme precipitation-high temperature events and assess their repercussions on the populace under three SSPs. The subsequent parts of this manuscript are organized as follows: Section 2 furnishes a comprehensive account of the utilized data and methodologies. The alterations in extreme events and population exposure are explored in Section 3. Following that, Sections 4 and 5 encapsulate the discourse and conclusion, respectively.

## Data and methods

Given China's vast expanse and diverse topography and climate types, this study employs the methodology proposed by Wang et al. (2020) to partition China into eight climatic sub-regions. These regions comprise the Western Arid-Semiarid (WAS) zone, Qinghai-Tibet (QT) region, Southern (S) zone, Central (C) zone, Northern (N) zone, Northeast (NE) zone, Southwest (SW) zone, and Eastern Arid (EA) zone (refer to Fig 1). These climatic zones are further categorized into three classes: arid regions (WAS and EA), transitional regions (NE, N, and QT), and humid regions (S, C, and SW). This classification framework facilitates a comprehensive analysis of the distribution of extreme climate events across both the entirety and specific sub-regions of China, enabling a nuanced examination of impact disparities.

### CMIP6 model output

This study utilizes climate scenario data from the CMIP6, specifically from Scenario MIP. CMIP6 represents an advancement over CMIP5, exhibiting enhanced capabilities in simulating regional temperatures and precipitation in China. It captures the nuanced climatic evolution of the region more effectively [29]. Initially, five global models that align with the research data requirements were selected for processing (see Table 1).

The five models were chosen for their robust global climate simulation capabilities, demonstrating accuracy in depicting historical temperature and precipitation trends, which proved effective in capturing regional climates, particularly in China and Asia [30]. Scholars have demonstrated the advantages of accurately assessing climate change in the study region by assembling these climate models and the certain advantages over this group have been shown [31].

Based on gridded observational data at an annual scale from 1970 to 2000, the paper employs statistical downscaling and Equidistant Cumulative Distribution Functions (EDCDF) for bias correction [32]. To facilitate overlay analysis with population data, multiple climate model datasets are uniformly resampled to a resolution of $0.5° \times 0.5°$.

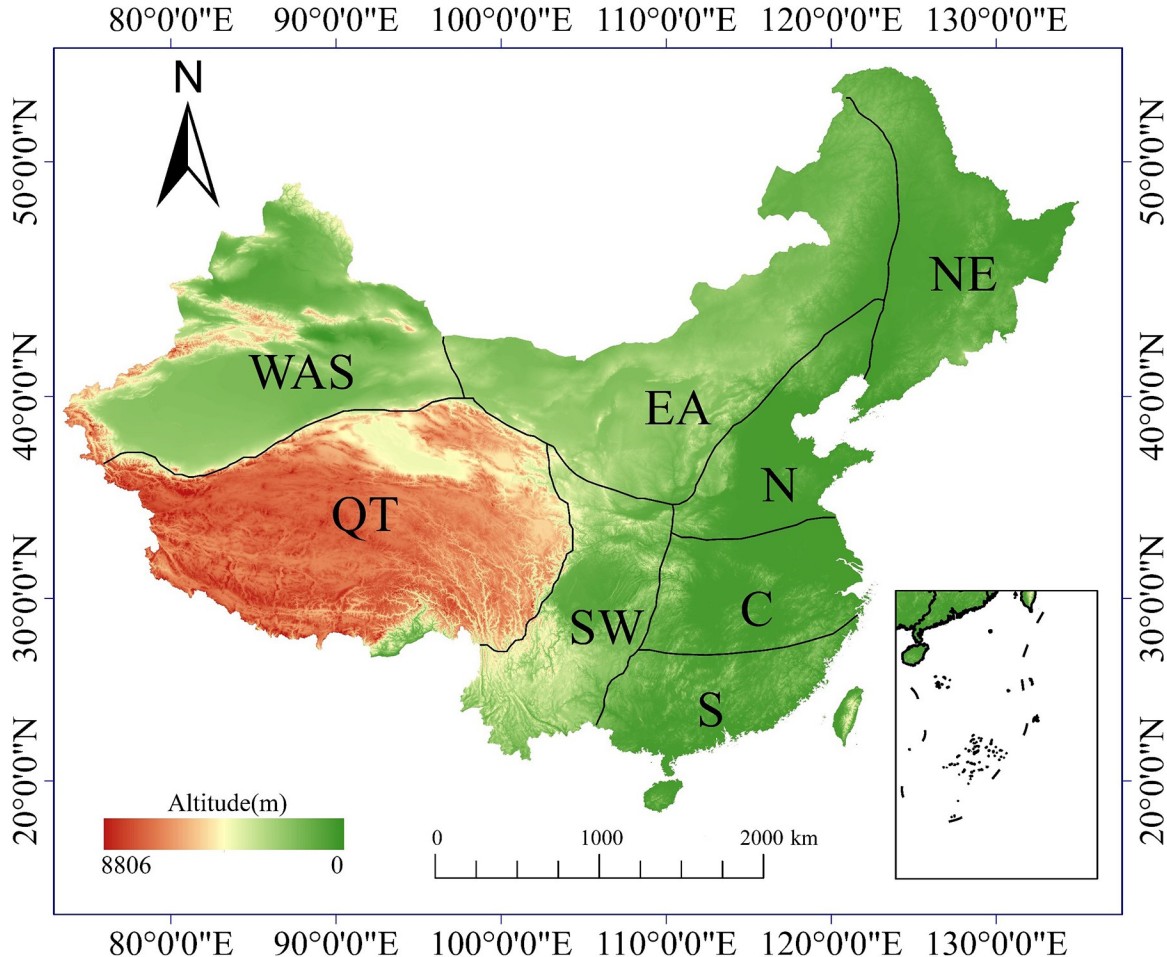

**Fig 1. Climatic sub-regions across China: Western arid and semi-arid region (WAS); Qinghai-Tibet region (QT); eastern arid region (EA); southwestern region (SW); northeastern region (NE); northern region (N); center region (C); southern region (S).** (The map is based on the standard map No. GS (2020)4619 downloaded from the website of the Standard Map Service of the Ministry of Natural Resources.).

## Observational meteorological data

Meteorological data utilized in this study were obtained from the China National Climate Center, China Meteorological Administration, through the China Surface Climate Data (Version 3.0). The dataset encompasses gridded daily maximum temperature and precipitation observations spanning the period from 1970 to 2000. The spatial resolution of this dataset is $0.5° \times 0.5°$, and it is accessible on the CMA Data Service Platform. The dataset comprises

**Table 1. CMIP6 models used in the study.**

| Model Name | Country | Research Institutions | Atmospheric Resolution |
|---|---|---|---|
| CNRM-ESM2-1 | French | CNRM, CERFACS | $1.406° \times 1.406°$ |
| CanESM5 | Canada | the Canadian Centre for Climate Modelling and Analysis | $1.406° \times 1.406°$ |
| IPSL-CM6A-LR | Europe | IPSL (Institute Pierre-Simon Laplace) | $1.259° \times 2.5°$ |
| MRI-ESM2-0 | Japan | MRI (Meteorological Research Institute | $1.125° \times 1.125°$ |
| MIRCO6 | Japan | Japanese Research Community | $1.406° \times 1.406°$ |

observations from 824 ground stations, covering the timeframe from January 1, 1951, to December 31, 2019.

## Population data

Population data for future China are derived from the Population and Economic Estimation Database under the Shared Socioeconomic Pathways (SSPs). This database incorporates data for China's regions, the Belt and Road Initiative region, and global projections, encompassing population by age, gender, education level, and GDP distributed across three industrial sectors. Three representative pathways, namely SSP2-4.5, SSP4-6.0, and SSP5-8.5, were utilized in this study. The data have a spatial resolution of $0.5° × 0.5°$ and were developed and made publicly accessible by Professor Tong Jiang's team at Nanjing University of Information Science and Technology. The dataset can be accessed through https://cstr.cn/31253.11.sciencedb.01683.

## Evaluation for performance of CMIP6

The Coupled Model Intercomparison Project Phase 6 (CMIP6), as the latest and most comprehensive phase of the CMIP initiative, facilitated the selection of five global climate models for evaluating their performance in simulating extreme climate events. Five indicators for precipitation and three for high temperatures were chosen to assess the models' capability in representing extreme climate conditions effectively, and the detailed information on these indicators can be obtained in Table 2.

## Taylor diagram

To assess the fidelity of the five CMIP6 models and the Multi-Model Ensemble (MME) in simulating precipitation and temperature in China, we employed the Taylor diagram method. The Taylor diagram provides a comprehensive and intuitive representation, summarizing the concordance between patterns exhibited by multiple models or simulations and those observed [33]. Quantification of similarity between the two involves assessing their correlation, root mean square differences, and variability, the latter expressed as standard deviation. The spatial correlation coefficient (SCC) between simulated and observed patterns is represented by the azimuthal position of the model point. The Standard Deviation Ratio (SDR) between simulation and observation is represented by the distance of the model point from the origin.

Table 2. Abbreviations, descriptive names, and definitions of precipitation and high temperatures.

| Types of indicators | Abbreviation | Descriptive Name | Definition | Unit |
|---|---|---|---|---|
| Precipitation | AP | Mean precipitation | Ratio of total annual precipitation to total number of days | mm/d |
| Precipitation | SDII | Precipitation intensity | Ratio of total annual precipitation to number of days with ≥0.1mm of daily precipitation | mm/d |
| Precipitation | APE | Precipitation frequency | Number of days with daily precipitation ≥ 0.1mm | d |
| Precipitation | EPI | Extreme precipitation intensity | Ratio of total precipitation to number of days with daily precipitation > 95% quantile | mm/d |
| Precipitation | R95P | Precipitation days exceeding 95th percentile | The number of days in the year when the daily precipitation exceeds the extreme precipitation threshold. | d |
| High temperature | ATI | Average temperature intensity | Ratio of annual cumulative maximum temperature values to total number of days | ˚C |
| High temperature | ETI | Extreme high temperature intensity | Mean temperature to those days with daily maximum temperature ≥ 35˚C | ˚C |
| High temperature | ETF | Extreme high temperature frequency | Number of days with daily maximum temperature ≥ 35˚C | d |

Normalized Root Mean Square Error (RMSE) signifies the bias between simulated and observed data. It is expressed as the distance from the model point to the reference point (REF, observed data point). The calculation formula is as follows.

$$\sigma_S = \sqrt[2]{\frac{1}{n}\sum_{i=1}^{n}(S_i - \bar{S})^2} \tag{1}$$

$$\sigma_O = \sqrt[2]{\frac{1}{n}\sum_{i=1}^{n}(O_i - \bar{O})^2} \tag{2}$$

Where, the $\sigma_S$ and $\sigma_0$ represent the standard deviations of the model field and the observed field, respectively. The $S$ and $O$ denote the model simulation index and the observed index for the climate system from 1970 to 2000, the $n$ represents the length of the time series, and the $i$ represents a specific grid point. The ratio of $S$ to $O$ is given by the standard deviation ratio (SDR), where $\sigma_S$ divided by $\sigma_0$ represents the ratio of the standard deviation of the simulation to that of the observation.

$$SCC = \frac{\frac{1}{n}\sum_{i=1}^{n}(S_i - \bar{S})(O_i - \bar{O})}{\sigma_S \sigma_0} \tag{3}$$

Where, the SCC (Spatial Correlation Coefficient) is the coefficient between the model field and the observed field in the two series of data.

$$RMSE = \sqrt{\frac{\sum_{i=1}^{n}(S_i - \bar{S})(O_i - \bar{O})}{n}} \tag{4}$$

Where, the RMSE (Root Mean Square Error) is the root mean square error between the model field and the observed field in the two sequences.

The SCC ranges from -1 to 1, with 1 indicating perfect positive correlation and -1 indicating perfect negative correlation. The SDR gauges the proportion of the model's standard deviation to observed values, with a value closer to 1 indicating smaller deviations. The RMSE measures the root mean square difference between the model and observed values; smaller RMSE values signify better model performance.

## Methodology for identifying compound events

This study focuses on extreme precipitation events, which are defined as instances significantly deviating from climatic averages, typically resulting in high impact despite their low probability of occurrence. To identify these events, a spatiotemporal composite method is employed, integrating extreme precipitation, temperature, and high temperature criteria. Individual extreme events are determined using percentile thresholds. Specifically, for each grid point within the defined spatiotemporal scope, daily maximum precipitation values are sorted, with the 95th percentile serving as the threshold for extreme precipitation events. This establishes threshold values for extreme precipitation across historical and future periods. High temperature is a critical component in identifying compound extreme events. A threshold is set, designating a day as potentially high-temperature when the maximum temperature exceeds 35°C. Compound extreme precipitation-high temperature events are identified when both conditions are met: the maximum temperature surpasses 35°C at a specific grid point, and there is precipitation exceeding the relative threshold within a 3-day window. To ensure accuracy, a time window concept is applied. Following the occurrence of high-temperature conditions, the subsequent 3 days are examined for precipitation exceeding the relative threshold. This

window accommodates potential lag effects between high temperature and precipitation, balancing continuity while avoiding a small sample size for quantitative

## Population exposure

The metric for population exposure is the count of individuals subjected to either extreme precipitation or extreme high temperature. The calculation involves multiplying the number of days with extreme precipitation or extreme high temperature by the respective population in each grid cell for a specific period. Exposure calculation is as follows:

$$\bar{E}_P = \frac{\sum_{n=1}^{30} T \times P}{30} \qquad (5)$$

where $\bar{E}_P$ represents 30-year averaged population exposure, $T$ and $P$ indicate the frequency of extreme events and population size, respectively, while $n$ is a year of the study period.

## Contributing factors to changes in population exposure

This study delves into the contributions of climate and population to overall exposure. The cumulative change in exposure is dissected into three components: climate, population, and their interaction effects. The methodology used for computing the relative significance of each effect aligns with that employed in [34]. It is calculated as follows.

$$\Delta E = T \times \Delta P + P \times \Delta T + \Delta P \times \Delta T \qquad (6)$$

Where $\Delta E$ is the change in population exposure, which compares the change in population exposure from the base period to two future periods: near-term future and long-term future. $T$ is the frequency of extreme events, $\Delta T$ is the change in frequency of extreme events, $P$ is the population size, and $\Delta P$ is the change in population size.

# Results

## Analysis of model performance

The mean values for each grid, covering both observations and historical modeled data from the base period (1970–2000), were computed. Following this, we evaluated how well five CMIP6 models simulated mean and extreme precipitation and temperature across China. We also assessed the multi-model ensemble (MME) using the Taylor diagram (Fig 2) for analysis. For precipitation indices, the SCC between all models and observations range from 0.42 to 0.97. Notably, the MRI-ESM2-0 and the MME exhibit SCC exceeding 0.80. The RMSE of simulations compared to observations range from 0.34 to 2.98. The SDR consistently exceeds 1, indicating favorable simulation performance of climate models for both the climatology and extremes of precipitation events. Consistent with this study, some researchers have observed an overestimation of spatial variability in precipitation frequency. This occurred when evaluating the performance of CMIP6 models in simulating precipitation across China. This overestimation may be associated with a positive bias in light precipitation frequency, suggesting that climate models simulate precipitation too frequently with insufficient intensity [35, 36].

For temperature indices, the SCC between all models and observations range from 0 to 0.98. Notably, the MRI-ESM2-0 and the MME exhibit SCC surpassing 0.86. The RMSE of simulations compared to observations ranges from 0 to 1.12. The SDR consistently exceeds 1, indicating satisfactory simulation performance of climate models for both the climatology and extremes of high-temperature events. Moreover, to conduct a more comprehensive assessment of CMIP6's simulation performance concerning precipitation and temperature, our analysis

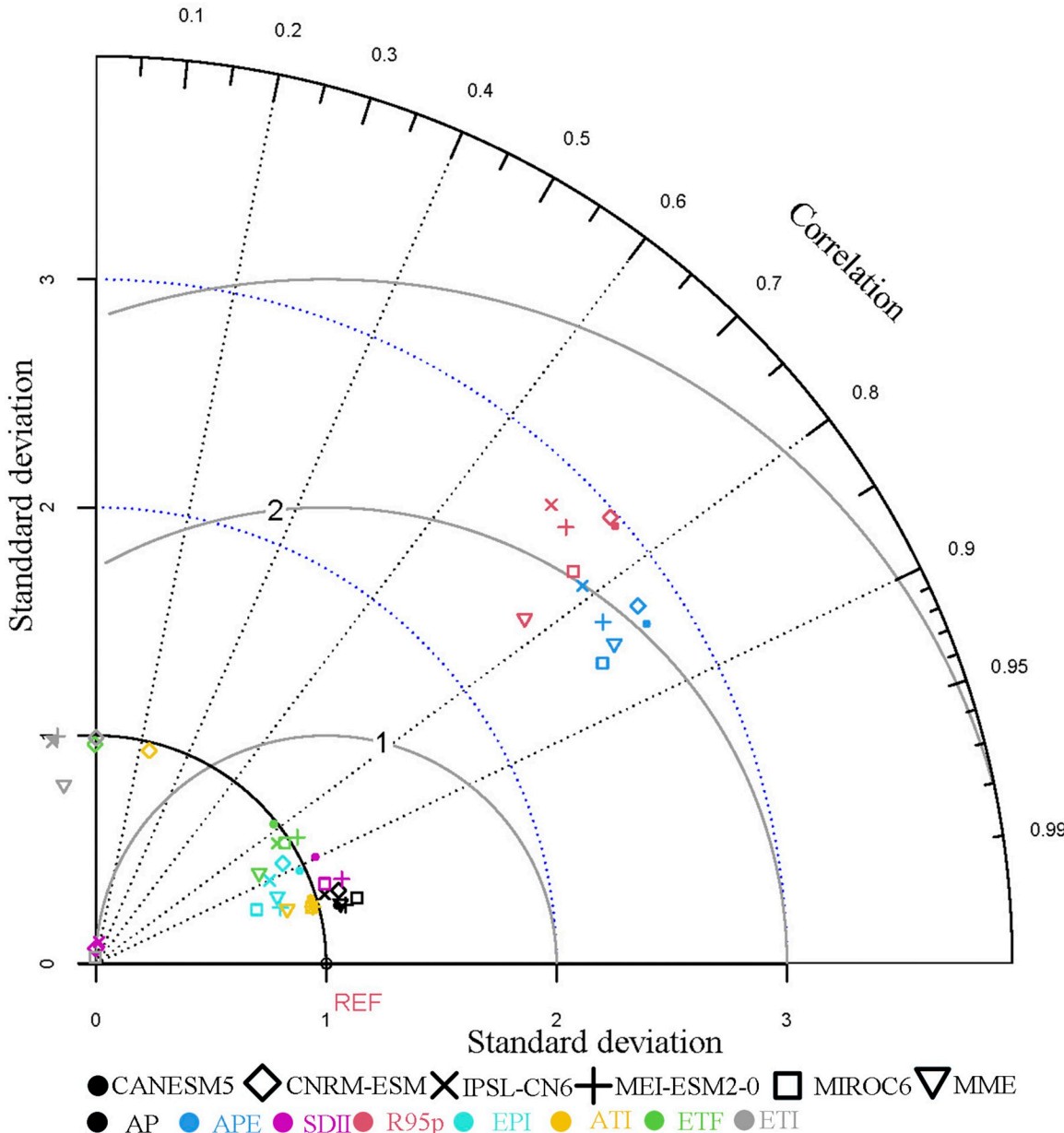

**Fig 2. Taylor diagrams of precipitation and high-temperature index simulations by 5 climate models and the Multi-Model Ensemble (MME) for China from 1970 to 2000.**

encompasses specific sub-regions (see S1 Fig). There are notable variations in performance among different models across these diverse sub-regions. Regarding precipitation indices, the Multi-Model Ensemble (MME) exhibits exceptional performance in sub-regions like EA, NE, QT, S, and SW, whereas the MRI-ESM2-0 excels in C and N. Conversely, for temperature indices, except for IPSL-CM6A-LR, which excels in SW, the MME consistently outperforms in all other sub-regions. In summary, the MME consistently showcases impressive simulation capabilities throughout China, validating its ability to accurately replicate precipitation and temperature patterns at both regional and subregional scales.

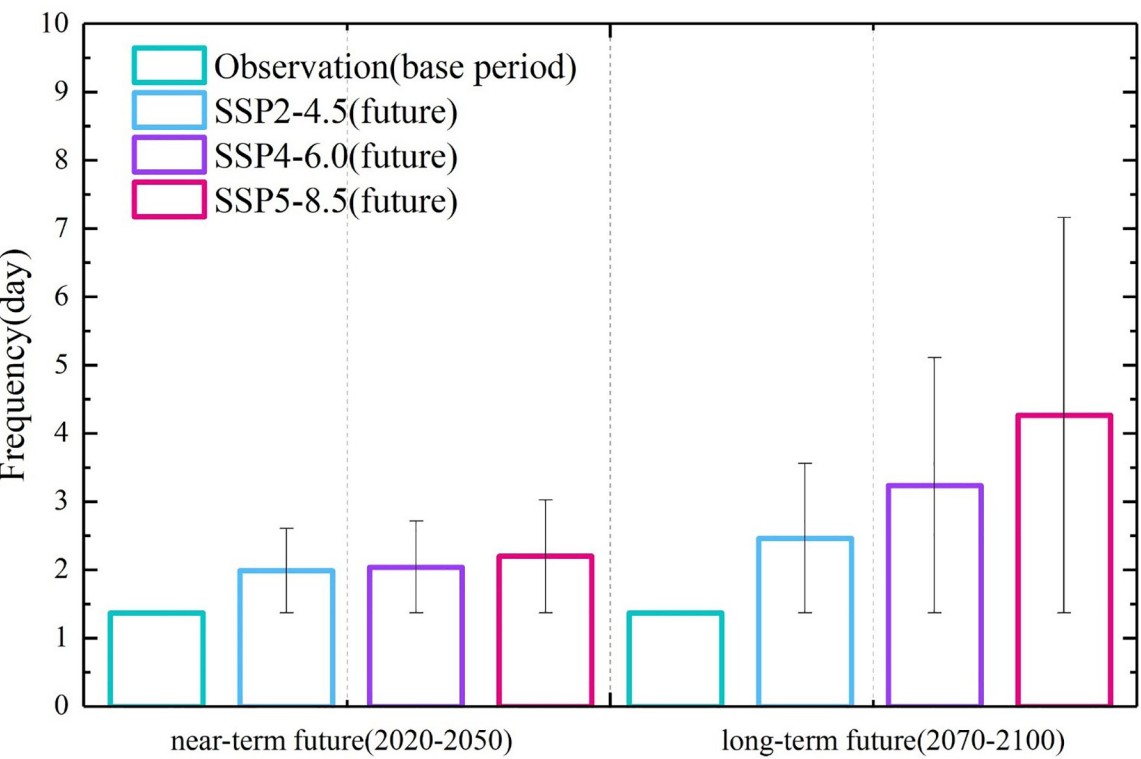

**Fig 3. The frequency of future compound extreme events in China for two periods (2020–2050 and 2070–2100) under historical conditions and three SSPs (SSP2-4.5, SSP4-6.0, SSP5-8.5).**

### Projection of compound events

As the Fig 3 show as, it is evident that during the baseline period (1970–2000), China's annual mean frequency of compound extreme events was 1.37 days per year. In the future projections, this frequency exhibits varying degrees of increase based on emission scenarios and time frames. During the near-term future (2020–2050), under different scenarios, the frequency of compound extreme events shows a gradual upward trend. Compared to the reference period, the frequency increases by 45.44% for SSP2-4.5, 49.18% for SSP4-6.0, and 60.62% for SSP5-8.5. By the end of the 21st century (2070–2100), the frequency of extreme events significantly rises under three SSPs. Relative to the baseline period, the frequency increases by 80.09% for SSP2-4.5, 136.55% for SSP4-6.0. The SSP5-8.5 exhibits the most substantial increase, reaching 4.26 days per year, a surge of 211.51% compared to the baseline period. This indicates a substantial rise in the frequency of compound extreme events in the future, particularly under high-emission SSPs, presenting a pronounced upward trend compared to historical levels.

The S2 Fig illustrates the temporal variations in the frequency of compound extreme events across eight climate sub-regions in China. The results reveal substantial diversity in the long-term trends of compound event frequency, with significant variations among different sub-regions. During the baseline period (1970–2000), the Northeast (NE), Southwest (SW), and Qinghai-Tibet Plateau (QT) regions all exhibited compound event frequencies. These frequencies were below 0.61 days/year, which is lower than the national average of 1.37 days/year. Moreover, these regions show relatively small changes compared to the baseline period, and there are minor differences among the SSPs simulated.

In stark contrast, the Northwest Arid Region (WAS) had a baseline compound event frequency of 4.82 days/year, and during the long-term future (2070–2100), under the SSP5-8.5,

the simulated frequency soared to 10.55 days/year. It's noteworthy that, despite having a higher frequency of extreme events than the national average, the change in this region is not substantial. The growth rate relative to the baseline period, the long-term future under the SSP5-8.5 is only 118.4%, which is lower than the national average of 211.51%.

## Spatial variations of compound events

Given the overall rise in the occurrence of compound extreme events anticipated in China's future, different climate regions show distinct responses to climate change. Investigating these differences provides more scientifically supported insights for regional climate adaptation and strategic adjustments. To explore these spatiotemporal differences and future spatial trends more precisely, Fig 4 presents the frequency (CE) and changes relative to the historical period

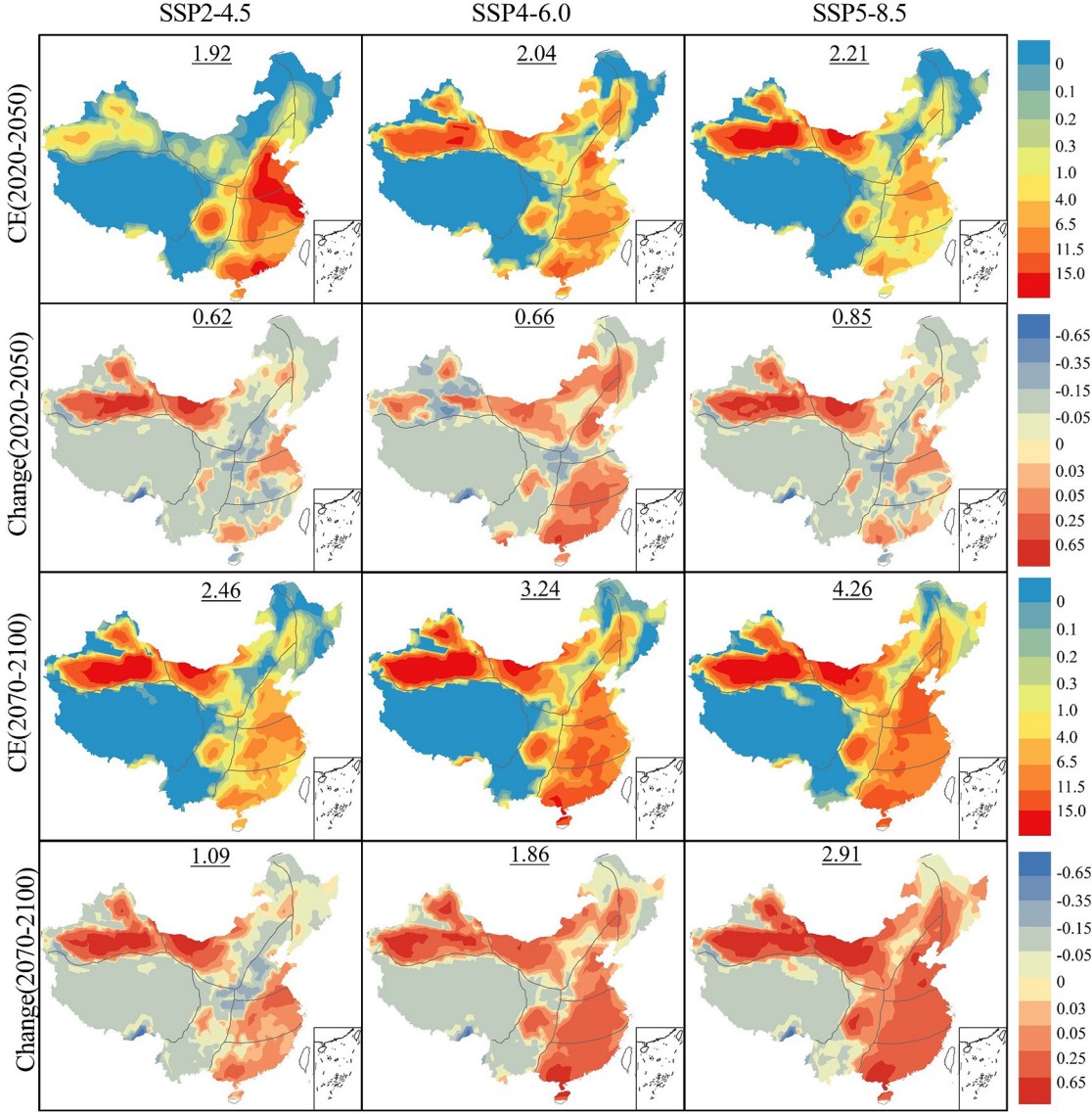

**Fig 4. Compared to period 1970–2000, future frequencies (rows 1 and 3) and changes (rows 2 and 4) in compound events in China for two future periods (2020–2050 and 2070–2100) under three SSPs (unit: D/y).** (The map is based on the standard map No. GS (2020)4619 downloaded from the website of the Standard Map Service of the Ministry of Natural Resources.).

of compound extreme events. This analysis covers two future periods (2020–2050, 2070–2100) under three SSPs. The numbers on each sub-figure represent the total differences in compound extreme events compared to baseline period.

The estimated spatial distribution of compound extreme event frequency in China reveals a multi-center pattern. During the long-term future (2070–2100), in the Northwest region simulated under the SSP5-8.5, including the Tarim Basin in Xinjiang and the Inner Mongolia Plateau, approximately 8–12 compound extreme events occur annually. In the eastern coastal regions, such as the Beijing-Tianjin-Hebei region and the Pearl River Delta, there are about 6–10 extreme high-temperature events annually, with some areas exceeding 10 occurrences. However, in several other climatic regions of China, the frequency of extreme compound events remains relatively low. For instance, in most parts of the Northeast and the vast Tibetan Plateau, only about 0.3 compound extreme events occur annually under the SSP5-8.5.

Compared to the baseline period, in near-term future (2020–2050), under all three SSPs, the China's annual compound extreme event frequency increases by 0.62 d/y (SSP2-4.5), 0.66 d/y (SSP4-6.0), and 0.85 d/y (SSP5-8.5). This increase, however, exhibits significant heterogeneity in space. Looking at the spatial trends, the growth in the Northwest region, North China Plain, and the Pearl River Delta far exceeds the national average, with an increase of about 0.5 d/y annually. In contrast, regions like the Central Plains, Northeast, and the Tibetan Plateau experience a reduction in the frequency of compound extreme events, decreasing by approximately 0.2 d/y relative to the baseline period.

By the end of the 21st century (2070–2100), except for a few regions like the Qinghai-Tibet Plateau and the Northeast of China, the entire study area will witness a widespread increase in the frequency of extreme events. Under all three SSPs, compared to the baseline period, the frequency of China's annual compound extreme events increases. Specifically, it increases by 2.46 d/y under SSP2-4.5, 3.24 d/y under SSP4-6.0, and 4.36 d/y under SSP5-8.5. Regions with a strong response to climate change, especially under high emission scenarios, show a substantial increase in the frequency of compound events. For example, in the South China region, where the baseline annual frequency is about 1.1 events, this value is projected to reach d/y by the end of the 21st century under the SSP5-8.5. This represents an increase of 553.04% compared to the baseline period. It is noteworthy that the Qinghai-Tibet Plateau does not show a clear increase in compound extreme events over time, and even experiences some reduction. By the end of the 21st century under the SSP5-8.5 scenario, a decrease of 49.55% compared to the baseline period suggests that this region is more stable or relatively less susceptible to the impacts of climate change.

## Population changes of China

The Fig 5 illustrates the population distribution in China under the SSP2-4.5, SSP4-6.0, and SSP5-8.5 across the near-term future (2020–2050) and long-term future (2070–2100). Overall, the future population trend in China indicates a decline in the total population and a high degree of urbanization. In all three SSPs, whether in near-term future or long-term future, there is a pronounced trend of population concentration in urban areas. Comparing population changes at different periods, relative to the baseline period (1970–2000), the population change in China is relatively small across near-term future. In the near-term future, the Chinese population is projected to increase only slightly compared to baseline period, with a potential decrease in population under the SSP4-6.0. However, by the end of the 21st century, China's population is expected to significantly decrease compared to the baseline period, with a more pronounced trend of population flow to large cities in the southeastern region of China. In this context, under the SSP2-4.5, the spatial distribution of China's population

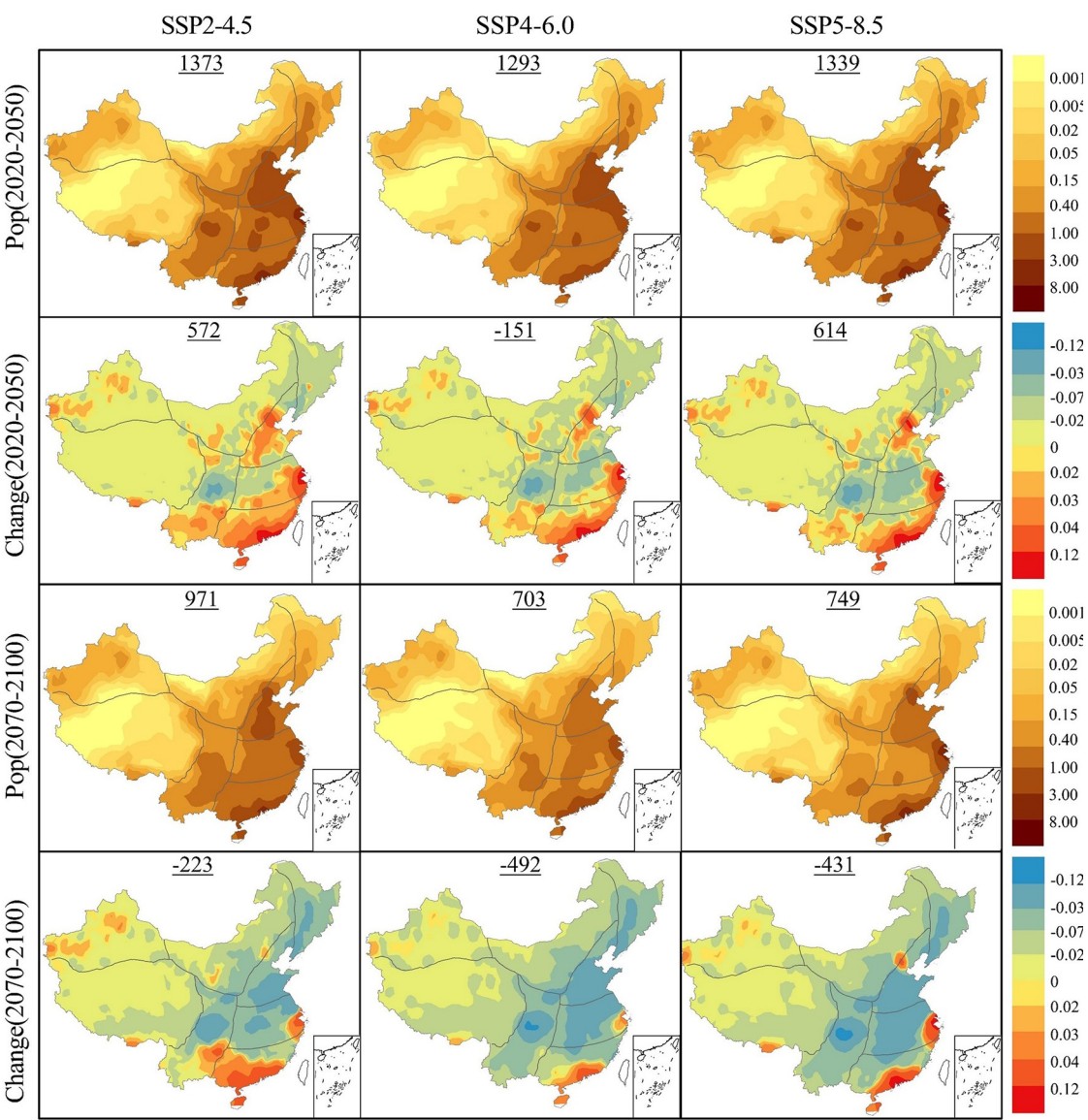

**Fig 5. Spatial distribution(brown) and changes (multi colour) in the population (unit: Million person) of China in two future periods (2020–2050, 2070–2100) under 3 SSPs (SSP2-4.5, SSP4-6.0, SSP5-8.5) compared to 1970–2000.** (The map is based on the standard map No. GS (2020)4619 downloaded from the website of the Standard Map Service of the Ministry of Natural Resources.).

exhibits relatively small urban-rural differences. However, under the SSP5-8.5, associated with higher greenhouse gas emissions, China's population is expected to show the highest degree of concentration. This is despite less population reduction compared to the SSP4-6.0. Furthermore, comparing SSP2-4.5 and SSP5-8.5, their differences in population distribution are limited, suggesting that climate change has a moderate impact on population distribution.

## Characteristics of population exposure

During the baseline period (1970–2000), the annual average population exposed within the impact range of compound extreme events in China was 1.979 billion people. This figure, compared to populations exposed to single extreme events (extreme precipitation and extreme

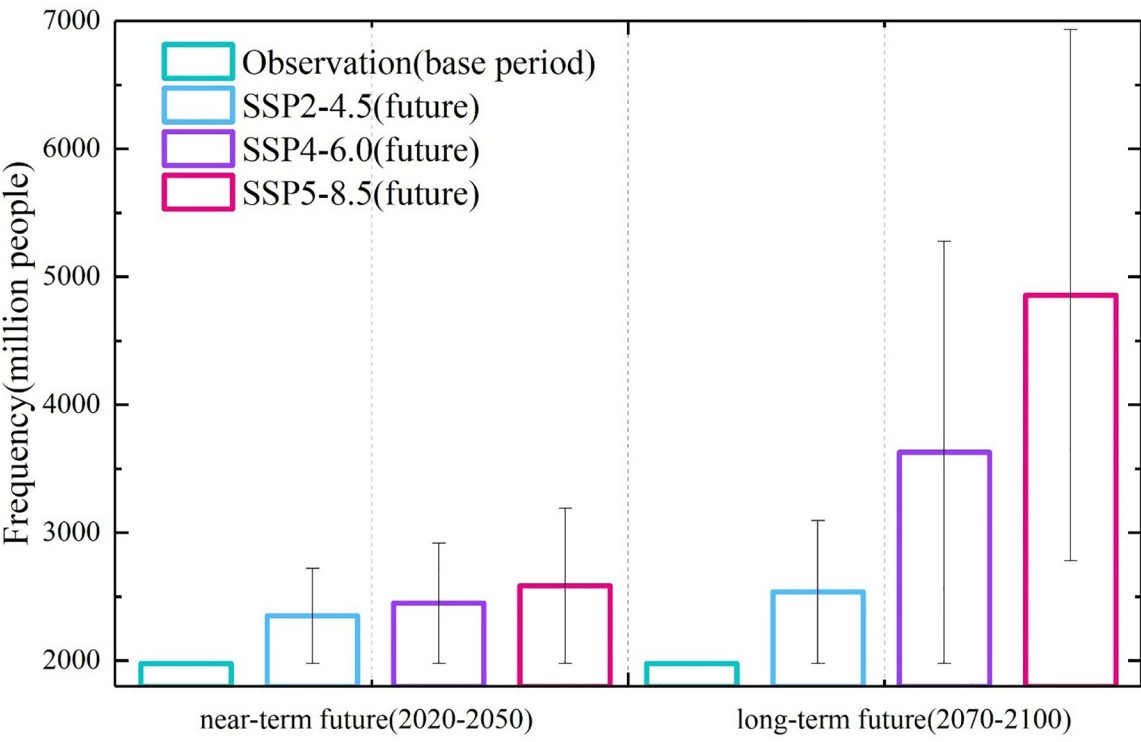

**Fig 6. The population exposed to compound events (unit: Million person) of China in baseline period (1970–2000) and two future periods (2020–2050, 2070–2100) under 3 SSPs (SSP2-4.5, SSP4-6.0, SSP5-8.5).**

high-temperature events), is relatively low, but the associated threats and risks still warrant attention. The annual cumulative population exposure to compound extreme events in the study region is illustrated in Fig 6. In the near-term future (2020–2050), the annual cumulative exposed populations under SSP2-4.5, SSP4-6.0, and SSP5-8.5 are approximately 2.335 billion, 2.449 billion, and 2.585 billion, respectively. These figures represent 1.71 times, 1.89 times, and 1.93 times the populations corresponding to the socio-economic pathways. Relative to the baseline period, the annual increase rates for each pathway are 0.12 billion, 0.18 billion, and 0.21 billion people, respectively.

Furthermore, by the end of the 21st century (2070–2100), the population exposed to compound events will further escalate under each emission pathway. The SSP2-4.5, SSP4-6.0, and SSP5-8.5 show an increase in exposed populations by 1.28 times, 1.83 times, and 2.45 times, respectively, relative to the baseline period. The annual cumulative exposed populations under these SSPs are projected to reach 2.583 billion, 3.629 billion, and 4.857 billion people. This indicates that in the two future periods, the population in China will face increasingly severe risks from compound extreme events, with this threat being more pronounced under high emission

The spatial distribution and relative changes in exposed population in the three SSPs for the two future periods (2020–2050 and 2070–2100) are depicted in Fig 7. Combining the information from Section 5.1, it is evident that China's population is most substantial under the SSP2-4.5, with the SSP4-8.5 exhibiting the most significant decline in total population. However, the trends in the exposed population to compound events demonstrate noteworthy differences. This disparity arises because the frequency of compound events in the SSP5-8.5 is markedly higher than in SSP2-4.5 and SSP4-8.5, resulting in SSP5-8.5 having the highest exposed

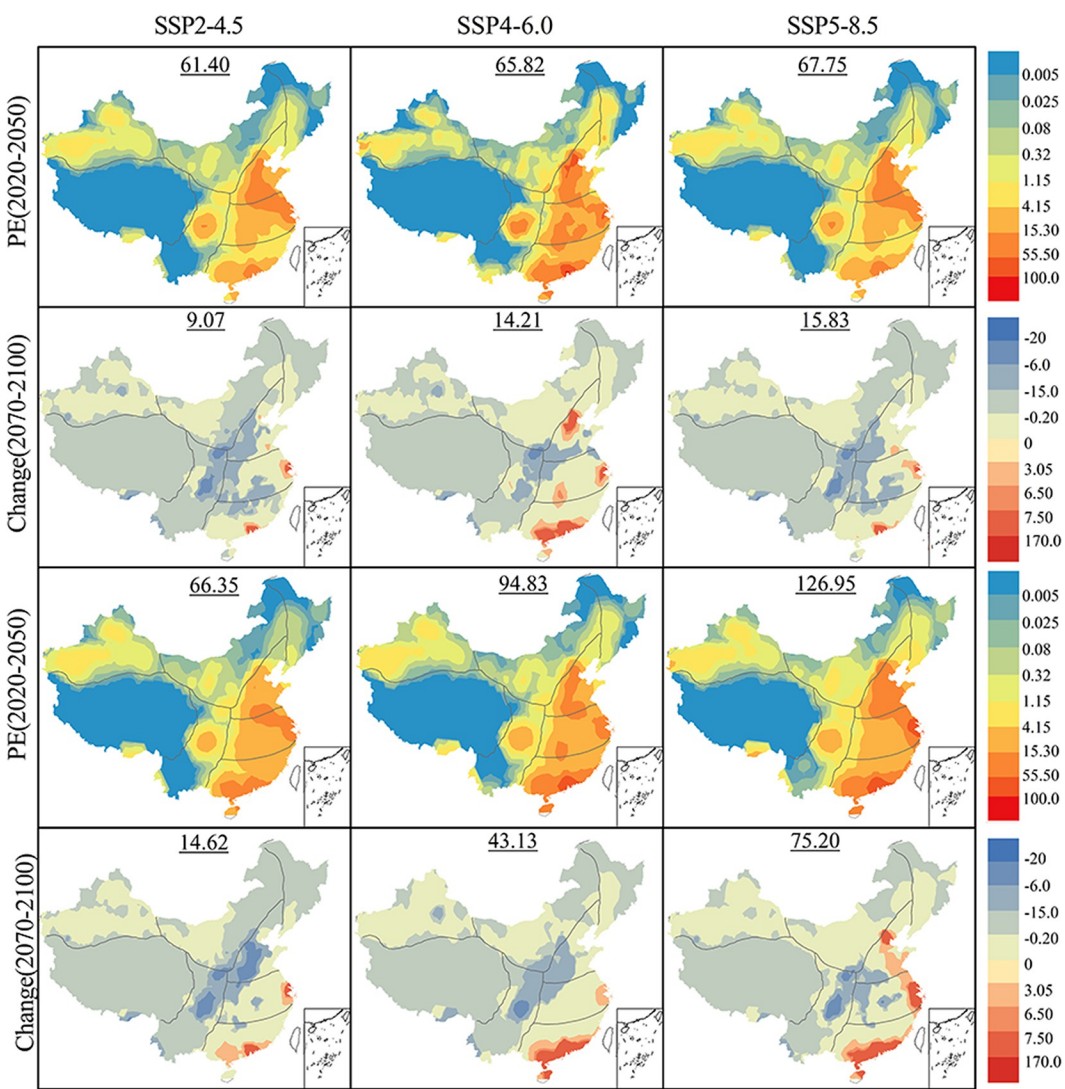

**Fig 7. Spatial distribution (the first and third rows) and changes (the second and fourth rows) of population exposed to compound events (unit: Million person) of China in two future periods (2020–2050, 2070–2100) under 3 SSPs (SSP2-4.5, SSP4-6.0, SSP5-8.5) compared to baseline period (1970–2000).** (The map is based on the standard map No. GS (2020) 4619 downloaded from the website of the Standard Map Service of the Ministry of Natural Resources.).

population. By the end of the 21st century, under this SSP, the annual average exposed population is 4.857 billion. This figure is approximately 5.2 times the total population under this pathway and 1.47 times the exposed population relative to the baseline period. Furthermore, although the SSP4-8.5 has a smaller total population compared to SSP2-4.5, under the SSP4-6.0, with increasing radiative forcing, the future occurrence of compound extreme events is notably higher than SSP2-4.5. Consequently, the exposed population under SSP4-8.5 experiences a higher growth rate than SSP2-4.5, especially in regions like the North China Plain and the Pearl River Delta. In these areas, the exposed population increases to an annual average of 25 million, with some areas exceeding 100 million, leading to severe exposure risks.

In terms of spatial distribution changes over the two future periods, in comparison to the baseline period, China's population exposed to compound extreme events exhibits significant imbalance. Although the total exposure increases continuously over the two future periods, a

decreasing trend is observed in the exposed population across the vast majority of the country. By the mid-21st century, major cities with concentrated populations in the simulated regions under all three SSPs, such as Shanghai, Beijing, Tianjin, Guangzhou, and Shenzhen, contribute significantly to the growth in exposed population, surpassing 55 million annually in all three SSPs.

Towards the end of the 21st century, the population exposed to compound events experiences a notable increase, and the high-value areas of exposed population expand further with the rise in radiative forcing. Under the SSP5-8.5, these high-value areas shift from highly populated cities like Beijing, Shanghai, and Guangzhou to the entire coastal region south of the Liaodong Peninsula. The annual average exposed population in these high-value areas under this SSPs increases by over 70 million relative to the baseline period. Although the regions with decreased exposed population experience a slight reduction, similar to the high-value areas, this trend intensifies with increasing radiative forcing. In the SSP5-8.5, some regions see an annual average reduction of approximately 12 million people relative to the baseline period.

## Factors influencing population exposure

The varying significance of three elements, climate, population, and their interaction, in influencing population exposure to extreme climate events across different future periods on a national scale is illustrated in Fig 8. Which results reveal a shift in the main driving factors affecting population exposure to compound extreme events in the two future periods. Specifically, in the near-term future, climate factors dominate, followed by population factors and climate-population interaction, with average proportions of 50.29%, 31.29%, and 18.42% across the three SSPs. During the long-term future, the population factors become the main factor

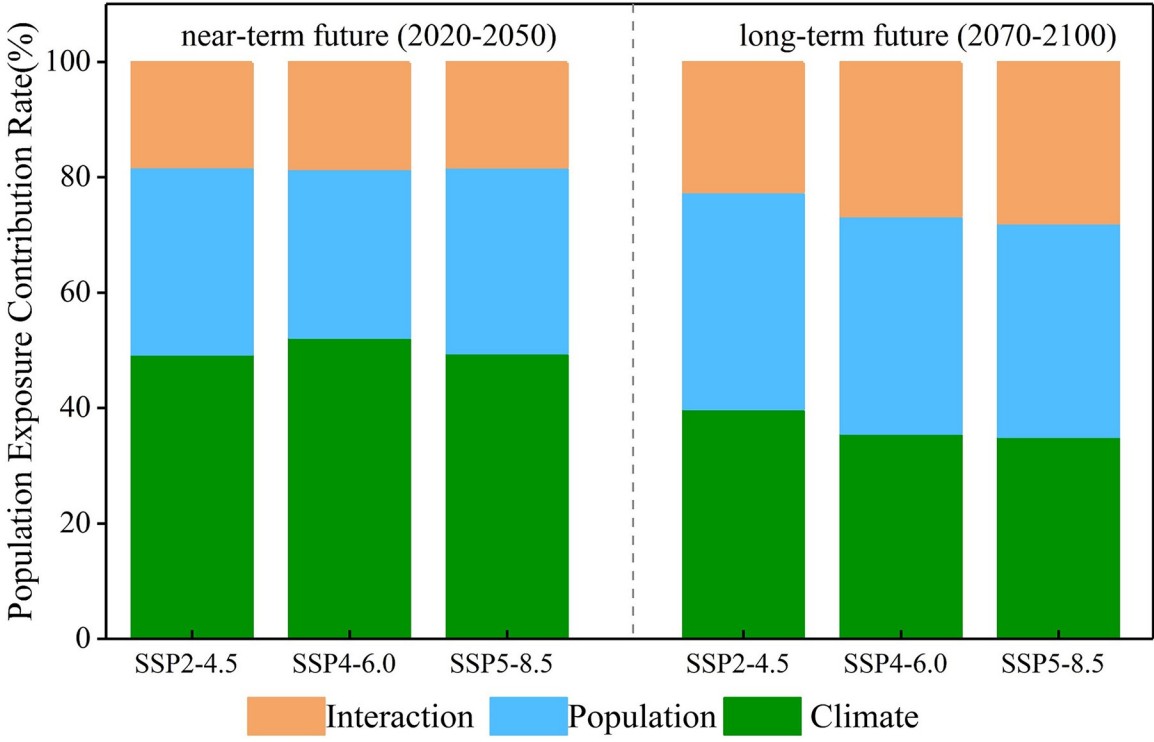

**Fig 8. Contributions from changes in population (green), climate (blue), and their interaction (orange) to population of China in two future periods (2020–2050, 2070–2100) under three SSPs (SSP2-4.5, SSP4-6.0, SSP5-8.5) (unit: %).**

driving of changes in population exposure to compound events. Predictions indicate a gradual reduction in the contribution of climate factors to future exposure changes, with average proportions decreasing to 37.92%. In contrast, the proportions for population factors and climate-population interaction increase to 38.60% and 23.48%, respectively, resulting in a more balanced distribution of the three influencing factors. It is noteworthy that, in the SSP2-4.5, climate factors still maintain a slight dominance. Contributions are 39.80% for climate factors, 37.58% for population factors, and 22.62% for climate-population interaction, respectively. This phenomenon is likely attributed to the relatively moderate population changes in SSP2-4.5 compared to the others, while compound event frequency variations remain pronounced.

From a spatial perspective, including the South (S), Central (C), North (N), and Northeast (NE) regions of China, the contribution rates of factors influencing population exposure to compound extreme events have undergone significant changes over time (see S3 Fig). Specifically, during the near-term future, been like to the overall trend in China, the climate factors are expected to have a significant impact on all hotspot regions. Over the near-term future, the population exposure in most climate regions is projected to be influenced by climate factors by more than 50%. Furthermore, with the strengthening of radiative forcing scenarios, the impact of climate factors is expected to intensify. For instance, in the N region, under the SSP5-8.5, the contribution rate of climate factors reaches 67.61%, making it the highest contributor nationwide. Under this SSPs, the contribution rates of population factors and population-climate interaction are 22.60% and 9.79%, respectively. In sharp contrast, the WAS region exhibits a more balanced contribution of the three factors, with population factors consistently being the dominant contributor to changes in population exposure to compound events, regardless of scenario. Under the SSP5-8.5, the contribution rates of climate factors, population factors, and population-climate interaction in this climate region are 34.03%, 41.98%, and 23.99%, respectively.

During the long-term future, a shift occurs in China, with climate factors gradually decreasing in proportion and population factors becoming increasingly dominant in driving changes in exposure to compound extreme events. The contribution rate of population-climate interaction also rises gradually. This shift is most pronounced in some regions such as the S, C, N, and NE, and it amplifies with increasing radiative forcing. For example, under the SSP5-8.5, the contribution rates of climate factors, population factors, and population-climate interaction in region C are 33.64%, 38.92%, and 27.44%, respectively. In comparison to the near-term future, the contribution rate of climate factors in this region decreases by 33.97%, while the contribution rates of population factors and population-climate interaction experience increases of 16.32% and 3.45%, respectively. Combining these findings with the earlier results suggests a significant correlation between this phenomenon and the changing spatial distribution of regional populations in the future.

## Discussions

Global warming has heightened the occurrence probability of extreme climate events worldwide, posing significant threats to human livelihoods and production [37, 38]. This study employs an innovative time-compounding approach to identify the compound extremes of high temperature and precipitation events. It investigates the future changes in extreme events and population dynamics in China, providing a comprehensive understanding of the impacts of compound extreme events. The spatial and temporal characteristics of extreme events across China and its sub-regions are analyzed, incorporating them into corresponding population exposure assessments to elucidate the spatial distribution changes and influencing factors across China's climate zones.

Our findings, based on the CMIP6 dataset, reveal pronounced changes in extreme precipitation and high temperatures. Compound extreme events are projected to increase significantly in the future, especially under the SSP5-8.5. Spatially, high-frequency occurrence centers of compound extreme events are identified, with regions like Northwestern China experiencing around 8–12 events annually. Conversely, extensive areas in Northeast China and the Tibetan Plateau exhibit lower frequencies, approximately 0.3 events annually, corroborating the findings of Zhang et al. [39].

This study foresees heightened risks from compound extreme events for China's population in the coming decades, particularly under high emission scenarios. Spatially, regions with high population exposure and growth concentrate in economically developed and densely populated areas, such as the Southeastern coastal and North China Plain regions. These areas face more frequent and intense compound extreme events, emerging as hotspots for future extreme precipitation and high-temperature impacts. Urbanization accelerates this vulnerability as the population becomes increasingly concentrated in urban centers over time.

Factors influencing population vulnerability to extreme events are investigated, emphasizing the interplay between population growth, climate, and their interactions [25]. Notably, climate change predominantly drives short-term exposure, constituting over 50% under SSP2-4.5 and SSP5-8.5. However, in the long term, this contribution decreases to below 30%, with population factors emerging as the dominant influence. The increasing significance of the interplay between population and climate is anticipated in molding patterns of exposure, particularly in China's Southern, Central, Northern, and Northeastern climate zones. The widespread impact of climate-related extreme weather underscores the critical need for precise climate predictions in China. Our study, focusing on sub-climates, offers significant advantages over broad climate-zone exposure assessments.

Acknowledging limitations, our study highlights discrepancies between CMIP6 model simulations and actual data, affecting the accuracy of population risk predictions. Future research should leverage higher-resolution regional climate models and diverse population datasets to enhance robustness. Additionally, exploring the impacts of hydrological and land-use changes on extreme events remains a crucial avenue for further investigation. Rigorous validation of our results is imperative for a comprehensive understanding of the complex interactions influencing population exposure to extreme events. In selecting the sample population, it was not subdivided, and in fact some studies have shown that the impact of extreme events will be more significant for people who are older or younger [34]. In addition, compound extreme events are not only harmful to the population, but also extremely costly to the socio-economy, which will be introduced in future studies.

## Conclusions

In this study, by utilizing data from the CMIP6, we project forthcoming climate change characteristics and delineate extreme events in China over two distinct time spans: the near-term future (2020–2050) and the long-term future (2070–2100). Subsequently, we integrate population projections with estimates of extreme event frequency under three different Shared Socioeconomic Pathways (SSPs) to quantitatively assess the population vulnerable to these impending extreme events. Finally, we conduct an in-depth analysis to identify the determinants of population exposure changes during extreme events, leading to the following prominent conclusions.

CMIP6 successfully reproduces spatial patterns of average precipitation and extreme precipitation (temperature) in China. The multimodel ensemble (MME) outperforms most individual models but exhibits significant biases across different subregions of China.

Additionally, precipitation frequency estimates show a systematic underestimation compared to actual observational results.

Examining extreme events in China across two temporal spans, we assess scenarios based on three Shared Socioeconomic Pathways (SSP): SSP2-4.5, SSP4-6.0, and SSP5-8.5. The findings reveal a general upswing in the frequency of compound extremes involving high temperatures and precipitation events. This is especially pronounced under conditions of high radiative forcing (RF), with the long-term future exhibiting a growth trend that exceeds that of the near-term future.

Regardless of scenario variations, high exposure regions and high growth regions during compound extreme events will concentrate in economically developed, densely populated areas such as China's southeastern coastal regions and the North China Plain. These areas will face the threat of more frequent and intense compound extreme events, becoming hotspots for future combined impacts of extreme precipitation and extreme high temperatures. In vast regions of other climate zones in China, a decrease in exposed population occurs due to factors like population loss and changes in compound events.

The frequency of extreme events and population size represent two crucial factors that significantly impact population exposure levels in extreme events. As time progresses, there is an anticipated shift in the dominant factor influencing population exposure to compound extreme events, transitioning from climate-related factors to population-related factors, especially concerning extreme high temperatures. The interplay between population dynamics and climate is poised to assume a progressively vital role in shaping the future trajectory of this exposure pattern.

## Supporting information

**S1 Fig. Taylor diagrams of precipitation and high-temperature index simulations by 5 climate models and the MME for 8 climate sub-regions of China from 1970 to 2000.**
(TIF)

**S2 Fig. The frequency of future compound extreme events in 8 climate sub-regions of China for two periods (2020–2050 and 2070–2100) under historical conditions and three SSPs (SSP2-4.5, SSP4-6.0, SSP5-8.5).**
(TIF)

**S3 Fig. Contributions from changes in population (green), climate (blue), and their interaction(orange) to population in of eight climatic sub-regions of China in the two future periods (2020–2050, 2070–2100) under 3 SSPs (SSP2-4.5, SSP4-6.0, SSP5-8.5).**
(TIF)

## Author Contributions

**Conceptualization:** Chao Gao.

**Formal analysis:** Ke Jin.

**Funding acquisition:** Yanwei Sun, Chao Gao.

**Investigation:** Ke Jin, Chao Gao.

**Methodology:** Ke Jin, Yanwei Sun.

**Project administration:** Xiaolin Sun, Yanwei Sun.

**Resources:** Ke Jin, Yanwei Sun.

**Software:** Xiaolin Sun.

**Validation:** Ke Jin, Yanwei Sun.

**Writing – original draft:** Ke Jin.

**Writing – review & editing:** Ke Jin, Yanjuan Wu.

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
