## [Decision Letter · Decision Letter 0]

28 Mar 2024

PONE-D-24-06038Spatial-temporal assessment of future population exposure to compound extreme precipitation-high temperature events across ChinaPLOS ONE

Dear Dr. Gao,

Thank you for submitting your manuscript to PLOS ONE. After careful consideration, we feel that it has merit but does not fully meet PLOS ONE’s publication criteria as it currently stands. Therefore, we invite you to submit a revised version of the manuscript that addresses the points raised during the review process.

We look forward to receiving your revised manuscript.

Kind regards,

Mohammed Magdy Hamed

Academic Editor

PLOS ONE

A clean copy of the edited manuscript (uploaded as the new *manuscript* file)”.

 [This research was supported by the Joint Funds of the Zhejiang Provincial Natural Science Foundation of China under Grant No. ZJMZ24D050014].  

5. We note that Figure 1, 4, 5 and 7 in your submission contain [map/satellite] images which may be copyrighted. All PLOS content is published under the Creative Commons Attribution License (CC BY 4.0), which means that the manuscript, images, and Supporting Information files will be freely available online, and any third party is permitted to access, download, copy, distribute, and use these materials in any way, even commercially, with proper attribution. For these reasons, we cannot publish previously copyrighted maps or satellite images created using proprietary data, such as Google software (Google Maps, Street View, and Earth). For more information, see our copyright guidelines: http://journals.plos.org/plosone/s/licenses-and-copyright.

a. You may seek permission from the original copyright holder of Figure 1, 4, 5 and 7 to publish the content specifically under the CC BY 4.0 license.  

6. We notice that your supplementary figures are included in the manuscript file. Please remove them and upload them with the file type 'Supporting Information'. Please ensure that each Supporting Information file has a legend listed in the manuscript after the references list.

Additional Editor Comments:

I noticed that reviewers advised to cite some publications. I advise you that any such requests are optional, and that only works crucial to the context of the manuscript should be considered for citation.

Reviewers' comments:

Reviewer's Responses to Questions

**Comments to the Author**

1. Is the manuscript technically sound, and do the data support the conclusions?

Reviewer #1: Yes

Reviewer #2: Yes

2. Has the statistical analysis been performed appropriately and rigorously? 

Reviewer #1: Yes

Reviewer #2: Yes

3. Have the authors made all data underlying the findings in their manuscript fully available?

Reviewer #1: Yes

Reviewer #2: Yes

4. Is the manuscript presented in an intelligible fashion and written in standard English?

Reviewer #1: Yes

Reviewer #2: Yes

5. Review Comments to the Author

Reviewer #1: Review comments on “Spatial-temporal assessment of future population exposure to compound extreme precipitation-high temperature events across China” by Jin et al. This manuscript used CMIP6 to predict the spatial-temporal variations of future population exposure to compound extreme events in China under three SSPs. In addition, this study showed the relative contribution of the factors contributing to changes in population exposure. The topic and the results are interesting. However, there are a few more details to be discussed in this manuscript. Therefore, I recommend a major revision. My specific comments are as follows:

(1)Page 2 The first sentence: You refer to IPCC AR6, the citation should more appropriately changed to 2021.

(2)Page 5: Why were these five models chosen and how did the other models in CMIP6 perform?

(3)Page 6: Would similar results be obtained using the other population data?

(4)Page 8: Definition of extreme precipitation thresholds not rigorous enough. Based on your description, you selected only the precipitation for the day in the study period (1970 - 2000) for ranking (the sample size is too small). You can expand the sample size by choosing 15 days of the study period, which corresponds to 7 days on either side of the target date.

(5)Page 10: Fig 2 Taylor diagram and the corresponding text description do not match, please re-check and confirm. For example: Page 10 Line 2-3, the SCC of precipitation indices between all models and observations range, what is represented on Fig 2 is not consistent with the range you stated. And “the MRI-ESM2-0 and the MME exhibit SCC exceeding 0.80” do not match either. Line 11-12, please check the range of RMSE, which is generally non-negative……

(6)Page 12 Fig 3: Please check why the observation values of the left and right sides of the figure are not consistent. In addition, the dashed lines on the left and right sides of the figure, which represent the SSP2-4.5 scenarios, are not of the same width. The same problem occurs in Fig 6.

(7)Page 15 Fig 4: Please check the text label on the left side of row 2, I guess you want to express "Change (2020-2050)". The same problem occurs in Fig 7.

Reviewer #2: The study titled “Spatial-temporal assessment of future population exposure to compound extreme precipitation-high temperature events across China” uses the outputs of multiple CMIP6 GCMs to estimate population exposure to compound temperature and precipitation extremes under three SSPs. The manuscript is well-written and presents interesting results, which would be of great interest to the readers. However, the following are some of my comments on the improvement of the manuscript, which I believe would improve its quality.

Comments:

Introduction:

• IPCC AR6 citation: IPCC 2012 >> IPCC 2022.

• The following papers are suggested for the authors to review for the introduction and discussion parts, as these papers also CMIP6 outputs for projections of independent and compound climate extremes in China and and neighboring regions. Compare and justify your results with the findings of these studies.

o https://doi.org/10.1016/j.scitotenv.2024.170133

o https://doi.org/10.1016/j.scitotenv.2023.162822

o https://doi.org/10.1016/j.atmosres.2023.106675

o https://doi.org/10.1088/1748-9326/ac9c70

• No keywords???

• The section “Method” should be “Data and Method” and then put the other sub-headings under this main heading.

Data:

• I am wondering why the authors chose only five CMIP6 GCMs while we have dozens of GCMs. Also, they didn’t use the newly released high-resolution, statistically downscaled, and bias-corrected outputs of the NEX-GDDP-CMIP6 models, which are highly recommended for climate change impact assessment at a regional scale.

• The authors used only three SSPs while overlooking SSP3, which is the most important one, particularly for estimating population exposure, as this SSP has been embedded with population effects. I would suggest to include SSP3 in the revision.

Method:

• Table 2 is also named as Table 1. Please correct it.

• Why do the authors evaluate the GCMs' performance by simulating daily independent extreme temperature and precipitation, but not their compound extremes? Any logic or science behind it?

• The definition of the compound extreme temperature and precipitation is not clear. Please rewrite it with more clear wording.

Results:

• In bar graph figures, the bars can be shown in different colors, instead of blank bars with dashed and line bars.

• Figures 5 & 7, please put the unit either with the color bars or in the caption.

• In limitations of the study, there should be some points on socioeconomic and demographic factors. See the following paper for reference.

o https://doi.org/10.1016/j.cliser.2022.100317

Discussion:

• The “Discussion” section can be extended a bit more, focusing on discussing/justifying the current results in light of the available literature.

• The following articles can be of good use for the authors to justify their results.

o https://doi.org/10.1016/j.wace.2023.100570

o https://doi.org/10.1029/2021EF002511

o https://doi.org/10.1029/2022EF003109

o https://doi.org/10.1016/j.atmosres.2022.106554

References

• The references list is incomplete and wrong. Some of the references are not cited correctly, i.e., ipcc >> IPCC

• Some are missing, i.e., (Shen et al., 2022)

• Likewise, many issues in the references.

6. PLOS authors have the option to publish the peer review history of their article (what does this mean?). If published, this will include your full peer review and any attached files.

Reviewer #1: No

Reviewer #2: No

---

## [Decision Letter · Decision Letter 1]

8 Jul 2024

Spatial-temporal assessment of future population exposure to compound extreme precipitation-high temperature events across China

PONE-D-24-06038R1

Dear Dr. Gao,

We’re pleased to inform you that your manuscript has been judged scientifically suitable for publication and will be formally accepted for publication once it meets all outstanding technical requirements.

Kind regards,

Mohammed Magdy Hamed

Academic Editor

PLOS ONE

Additional Editor Comments (optional):

Reviewers' comments:

Reviewer's Responses to Questions

**Comments to the Author**

1. If the authors have adequately addressed your comments raised in a previous round of review and you feel that this manuscript is now acceptable for publication, you may indicate that here to bypass the “Comments to the Author” section, enter your conflict of interest statement in the “Confidential to Editor” section, and submit your "Accept" recommendation.

Reviewer #1: All comments have been addressed

Reviewer #2: All comments have been addressed

2. Is the manuscript technically sound, and do the data support the conclusions?

Reviewer #1: Yes

Reviewer #2: Yes

3. Has the statistical analysis been performed appropriately and rigorously? 

Reviewer #1: Yes

Reviewer #2: Yes

4. Have the authors made all data underlying the findings in their manuscript fully available?

Reviewer #1: Yes

Reviewer #2: Yes

5. Is the manuscript presented in an intelligible fashion and written in standard English?

Reviewer #1: Yes

Reviewer #2: Yes

6. Review Comments to the Author

Reviewer #1: The authors have well revised the manuscript following my previous comments. I have no further suggestions.

Reviewer #2: The authors have addressed all my comments in the revised manuscript, and I now recommend it for publication in PLOS One.

7. PLOS authors have the option to publish the peer review history of their article (what does this mean?). If published, this will include your full peer review and any attached files.

Reviewer #1: No

Reviewer #2: No

---

## [Editor Report · Acceptance letter]

23 Jul 2024

PONE-D-24-06038R1 

PLOS ONE

Dear Dr. Gao, 

I'm pleased to inform you that your manuscript has been deemed suitable for publication in PLOS ONE. Congratulations! Your manuscript is now being handed over to our production team.

Kind regards, 

on behalf of

Dr. Mohammed Magdy Hamed 

Academic Editor

PLOS ONE